# An Alternative Depressant of Chalcopyrite in Cu–Mo Differential Flotation and Its Interaction Mechanism

**Xuemin Qiu *, Hongying Yang *, Guobao Chen** 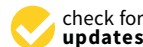 **and Wenjie Luo**

School of Metallurgy, Northeastern University, Shenyang 110819, China; chengb@smm.neu.edu.cn (G.C.); lwj66463863@126.com (W.L.)

* Correspondence: qxm20080902@163.com (X.Q.); yanghy@smm.neu.edu.cn (H.Y.);
  Tel.: +86-024-8367-2932 (X.Q.); +86-024-8367-3932 (H.Y.)

**Abstract:** Carboxymethylcellulose (CMC) is a nontoxic and biodegradable polysaccharide, which can potentially replace the frequently used hazardous depressants in Cu–Mo separation. However, a lack of understanding of the interaction mechanism between the CMC and the minerals has hindered its application. In the present study, it is found that $50 \text{ mg} \cdot \text{L}^{-1}$ CMC can inhibit chalcopyrite entirely in the pH range 4–6, while having little effect on molybdenite. The results also showed that the inhibition effect of the depressant for chalcopyrite enhanced with the increase of the degree of substitution (DS) and molecular weight ($M_w$) of CMC. The low DS and high $M_w$ of CMC were detrimental to the Cu–Mo separation flotation. Furthermore, CMC adsorption was found to be favored by a positive zeta potential but hindered by the protonation of the carboxyl groups. An electrochemical study showed that CMC inhibited 92.9% of the electrochemical reaction sites of chalcopyrite and greatly reduced the production of hydrophobic substances. The XPS and FTIR measurements displayed that the chemisorption was mainly caused by $Fe^{3+}$ on the chalcopyrite surface and the carboxyl groups in the CMC molecular structure.

**Keywords:** chalcopyrite; CMC; depressant; flotation; Cu–Mo separation; surface chemistry

## 1. Introduction

Molybdenum (Mo) is one of the most important rare metals. It is a widely used additive in steel alloys, high-temperature alloys, and other materials [1,2]. Mo usually occurs as molybdenite ($MoS_2$) and is usually associated with copper sulfide ores such as chalcopyrite ($CuFeS_2$) [3,4]. Mo concentrate is commonly obtained as a by-product from copper ore by a two-stage froth flotation technique [5]. The first stage produces a copper and molybdenum bulk concentrate, and the second stage separates Mo from the copper sulfide by selective flotation. As molybdenite is inherently more floatable than chalcopyrite [6], chalcopyrite is inhibited by a depressant that selectively hinders the collector adsorption to chalcopyrite or increases the hydrophilicity of chalcopyrite during the separation. The most routinely used chalcopyrite depressants are inorganic reagents such as sodium sulfide [7–10], sodium thioglycolate [11,12] and sodium cyanide [13]. These reagents are extremely toxic and applied in high amounts, which are harmful to both humans and the environment. Owing to the increasingly stringent control regulations against environmental pollution, the need to replace such toxic and hazardous depressants with more environment-friendly chemicals is gaining urgency.

Naturally occurring polysaccharides such as starch, dextrin, cellulose, and guar gum have been reported as candidate selective depressants in the recovery of valuable metals from bulk concentration [14,15]. These polymers are not only nontoxic but also biodegradable and relatively inexpensive. In numerous laboratory investigations, sulfide minerals such as galena [16–18] and pyrite [19–22] have been successfully depressed. Many studies have considered the feasibility of

polysaccharides as a chalcopyrite depressant in Cu–Mo separation flotation, including ATDT [23], DMTC [24], tannin [25], chitosan [26], and so on. Carboxymethylcellulose (CMC), a natural polysaccharide which is widely used as a talc depressant and food additive [27–29], might be suitable for this purpose. CMC is known to inhibit chalcopyrite under certain conditions [30,31]. However, the interaction mechanism between CMC and the chalcopyrite surface has rarely been investigated. The general interaction between polysaccharides and sulfide ores is highly controversial and needs resolving in many further studies. The present study assessed the efficiency of CMC as a depressant of chalcopyrite in Cu–Mo differential flotation and elucidated the underlying interaction mechanism.

## 2. Materials and Methods

### 2.1. Materials

Figure 1a shows the molecular structure of CMC. The molecular weight ($M_w$) and degree of substitution (DS) of this common polysaccharide are 50,000–300,000 and 0.5–1.5, respectively. CMC was obtained from Sinopharm Chemical Reagent Co., Ltd. (Shanghai, China). Sodium ethyl-xanthate (SEX), used as a collector, was obtained from Tieling Flotation Reagent Co., Ltd. (Liaoning, China). Pure mineral samples of chalcopyrite and molybdenite were obtained from Taobao. The samples were crushed, handpicked, and dry-ground in a porcelain ball mill (GCHQM-2L, Nanjing, China). The X-ray fluorescence (Shimadzu-XRF1800, Shimadzu, Kyoto, Japan) and X-ray powder diffraction analysis (Bruker-D8 Discover, Bruker, Mannheim, Germany) results are shown in Table 1 and Figure 1b, respectively.

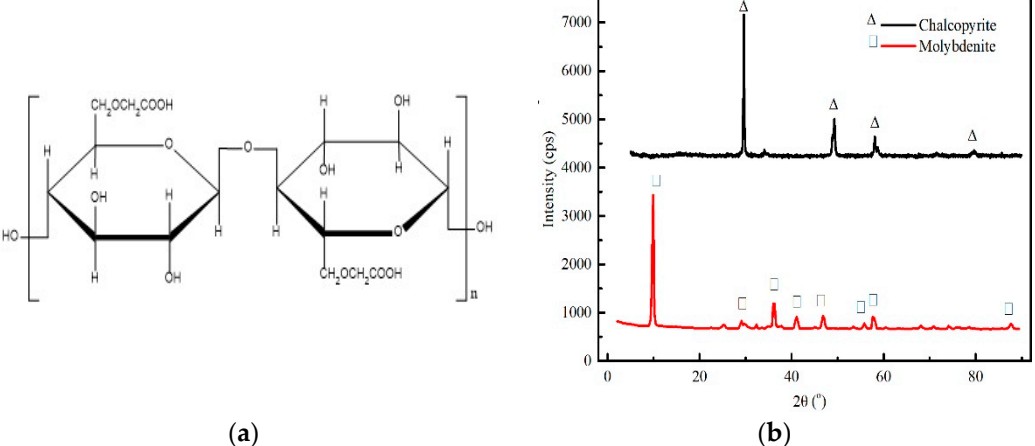

(a)                                                                                                         (b)

**Figure 1.** (**a**) Molecular structure of CMC; (**b**) X-ray diffraction spectra of chalcopyrite and molybdenite.

**Table 1.** X-ray fluorescence results of chalcopyrite and molybdenite (mass fraction, %).

| Element | Cu | Fe | S | Mo | Si | O |
|---|---|---|---|---|---|---|
| Chalcopyrite | 33.7 | 32.3 | 32.2 | | 0.6 | 1.2 |
| Molybdenite | | | 40.2 | 58.8 | 0.5 | 0.5 |

### 2.2. Micro-Flotation Experiment

In order to accurately study the effects of CMC on chalcopyrite and molybdenite, pure minerals were used. The floatabilities of the single and mixed minerals were measured in a suspended flotation cell (XFGCII5-35, Jinlin Exploration Machinery Plant, Changchun, China). The mass fraction of particle size less than 74 μm of chalcopyrite and molybdenite were 82.4% and 78.7%, respectively. The mineral suspension was prepared by adding 25 g of mineral powder to 100 mL of solution while using a stirrer to agitate. The pH of the mineral suspension was first adjusted to the desired value (2, 4, 6, 8, 10, 12) with concentrated NaOH or $H_2SO_4$ stock solution. The prepared CMC solution was added to the

desired concentration while conditioned for 5 min. The flotation was initiated by air inflation at a flow rate of 100 mL·min$^{-1}$ for 3 min.

### 2.3. Adsorption of CMC on Chalcopyrite and Molybdenite

Adsorption measurements were conducted using the batch depletion method. A particle size of less than 38 μm was used for adsorption tests. A 10 wt % pure mineral suspension was prepared and separated into individual vials. Solutions of the same polymer dosage (10 wt %) were added to mineral suspensions of different pH values (2–12, in 1-unit increments). The resulting suspensions were tumbled for 20 min (simulated the process of flotation). After tumbling, the suspensions were centrifuged at, and the polymer concentration of the supernatant was determined by a total carbon analyzer (Shimadzu TOC-V, Shimadzu, Kyoto, Japan).

### 2.4. Zeta Potential Measurements

The zeta potential of the mineral samples was measured using an electrophoresis instrument (POWEREACH JC94H2, Xiamen McLaren Jingruike Instrument Co., Ltd., Xiamen, China). The samples were ground to 2 μm-diameter particles under an inert gas with a ball mill (Alc Minerals Technology 1.5 L, Alc Minerals Technology Co., Ltd., Jinhua, China). In each test, 50 mg of the mineral was added to a beaker containing 50 mL of 0.1 mol·L$^{-1}$ KCl solution and the suspension was stirred for 5 min. The pH of the suspension was adjusted to the desired value using 0.1 mol·L$^{-1}$ $H_2SO_4$ and 0.1 mol·L$^{-1}$ NaOH.

### 2.5. Electrochemical Tests

Electrochemical measurements were performed using a conventional three-electrode system. The reference, counter, and working electrodes were a saturated calomel electrode, a 1-cm$^2$ platinum-foil electrode, and a mineral electrode, respectively. Prior to each test, the surface of the chalcopyrite electrode was polished dry with a 5000-grit diamond-carbide paper. The effect of CMC on chalcopyrite was evaluated by cyclic voltammetry and a Tafel plot.

### 2.6. Infrared Spectrum Measurements

Single minerals (1.0 g) were added to the desired amount of solution and magnetically stirred for 30 min. After settling on the laboratory bench for another 30 min, the samples were added to the differential pH solutions and CMC and then filtered. The solid was obtained using a vacuum-drying method and subjected to infrared spectrum measurements by Fourier-transform infrared (FTIR) spectrometry.

### 2.7. X-ray Photoelectron Spectroscopy Measurements

In preparation for X-ray photoelectron spectroscopy (XPS) analysis, chalcopyrite (5 g) was mixed with 50 mg·L$^{-1}$ CMC in a 200 mL beaker at pH 5. The slurry was filtered, conditioned in an incubator at 25 °C for 30 min, and then washed with 200 mL distilled water. Finally, the sample was dried in a vacuum desiccator and subjected to XPS analysis. All XPS measurements were conducted within 12 h of the sample preparation to minimize oxidation.

## 3. Results and Discussion

### 3.1. Micro-Flotation of Single Mineral Tests

In the absence of CMC, the recoveries of chalcopyrite and molybdenite exceeded 80% (Figure 2a). Both minerals were easily recovered by flotation in xanthate solution. The recovery was slightly decreased at pH below 4 because xanthate decomposes under very acidic conditions. The addition of 50 mg·L$^{-1}$ CMC completely inhibited chalcopyrite floatability in the 4–6 pH range. Outside of this range, the chalcopyrite recovery rapidly increased. In all pH tests, CMC addition slightly affected

molybdenite flotation. The results confirm the feasibility of CMC as a depressant of chalcopyrite in Cu–Mo differential flotation.

### 3.2. The Effect of DS and Mw

The important parameters of CMC are the degree of substitution (DS) and the molecular weight ($M_w$). As shown in Figure 2b, increasing the DS and $M_w$ improved the inhibition of CMC to chalcopyrite flotation. When DS was 1.5 and $M_w$ was 300,000, the chalcopyrite recovery was 11.3%. The molybdenite recovery increased with increasing DS and decreased with increasing $M_w$. Low DS and high $M_w$ degraded molybdenite flotation, whereas high DS and high *Mw* favored chalcopyrite inhibition. To optimize the separation, CMC with DS = 1.5 and $M_w$ = 200,000 was selected as the depressant of Cu–Mo differential flotation.

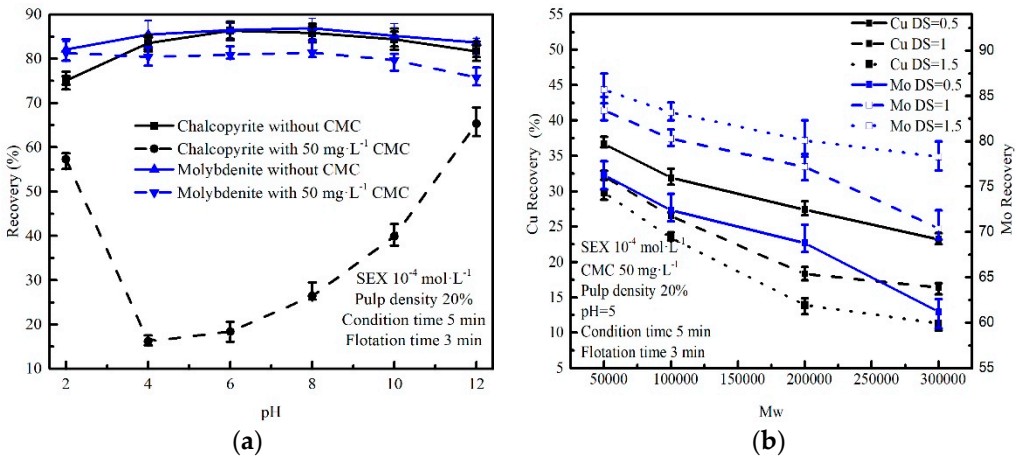

**Figure 2.** (**a**) Effects of pH and (**b**) molecular weight (with varying degree of substitution) on chalcopyrite and molybdenite flotation in the CMC system.

### 3.3. Flotation of Chalcopyrite–Molybdenite Mixed Minerals

Figure 3a shows the flotation test results of the artificial chalcopyrite–molybdenite mixture. The recovery and grade of the copper decreased as the CMC concentration increased up to 60 mg·L$^{-1}$. Above this level, the copper recovery and grade were slightly affected by CMC concentration. Molybdenite was not inhibited by CMC, except at high concentrations, where a slight decrease was noted. The Mo grade was also slightly decreased at CMC concentrations above 60 mg·L$^{-1}$. At high concentrations, CMC might increase pulp viscosity [32]. Some molybdenite remained as tailings and some chalcopyrite was mixed in the concentrate. The best CMC concentration for separating Cu and Mo was 60 mg·L$^{-1}$. The recovery and grade of Mo were 80% and 44.7%, respectively.

### 3.4. Adsorption of CMC to Chalcopyrite and Molybdenite

Figure 3b plots the adsorption densities of CMC on chalcopyrite and molybdenite as functions of pH. The CMC was weakly adsorbed to molybdenite (<1 mg·m$^{-2}$) at pH values less than 10. At higher pH, the adsorption was slightly increased (to nearly 2 mg·m$^{-2}$) possibly because the surface oxidation favored CMC adsorption to molybdenite. CMC was strongly adsorbed to chalcopyrite (adsorption density > 15 mg·m$^{-2}$) in the pH range 4–6. The adsorption density decreased at pH values below 4. The carboxyl groups in the CMC were the anchor points for the adsorption. The CMC molecule partially protonates at pH below 4 (–COO$^-$ to –COOH) and is fully protonated below pH 2 [33]. This protonation likely contributed to the decrease in adsorption at a very acidic pH value [34]. The adsorption density also decreased at pH values above 6. Many previous studies suggested that zeta potential was changed.

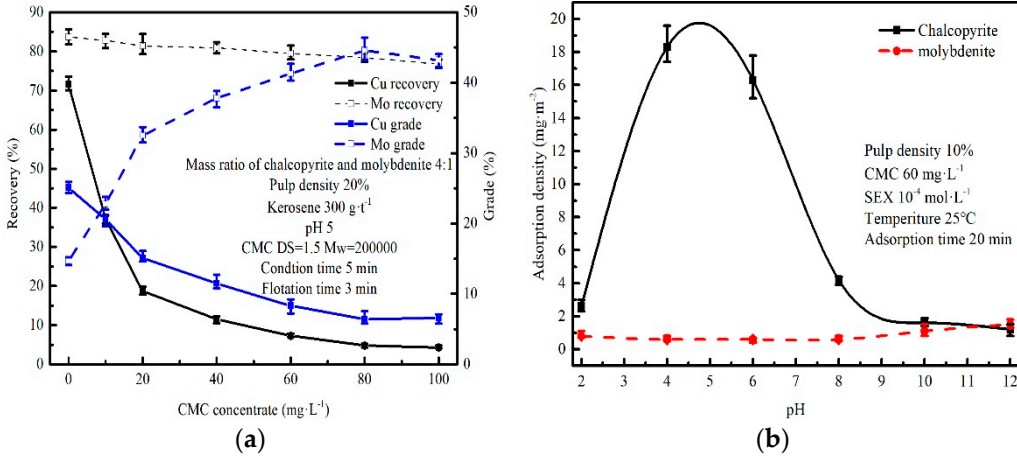

**Figure 3.** (**a**) Effect of CMC concentration on the Cu–Mo mixed minerals separation flotation; (**b**) adsorption of CMC on the chalcopyrite and molybdenite.

## 3.5. Zeta Potential Measurements

The zeta potentials of chalcopyrite in the presence and absence of CMC were plotted as functions of pH in Figure 4a. The species of Fe (Figure 5a) and Cu (Figure 5b) in solution were calculated using the CHEAQS program (P2017.3, Wilko Verweij, Amsterdam, Netherlands). The zeta potential of chalcopyrite became more negative as the pH increased from 3 to 11. In the absence of CMC, the isoelectric point (IEP) was between pH 6 and pH 7, consistent with the measurements of other researchers [35,36]. In the presence of 50 mg·L$^{-1}$ CMC, the zeta potential of chalcopyrite decreased from that of the no-CMC case (especially in the 4–8 pH range), indicating that many CMC molecules were adsorbed to the surfaces of chalcopyrite particles. This can be explained by the many carboxylic groups in the CMC molecular structure at pH above 3.8 [37]. Electrostatic interaction took an important role in the CMC adsorption. According to the CHEAQS calculation, iron cations appeared at pH values below 8 and dominated at pH values below 6. Meanwhile, copper cations appeared at pH levels below 12 and dominated at pH values below 10.5. The CMC carboxylic groups reacted with the cations on the chalcopyrite surface by electrostatic interaction, as shown in Figure 4b.

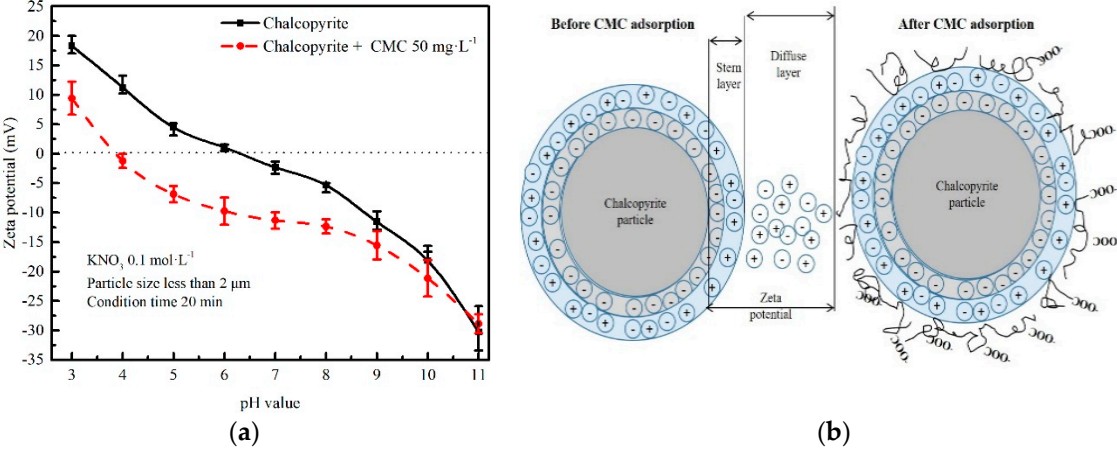

**Figure 4.** (**a**) pH dependence of the zeta potential of chalcopyrite in the absence and presence of CMC; (**b**) electrostatic adsorption mechanism of CMC on chalcopyrite.

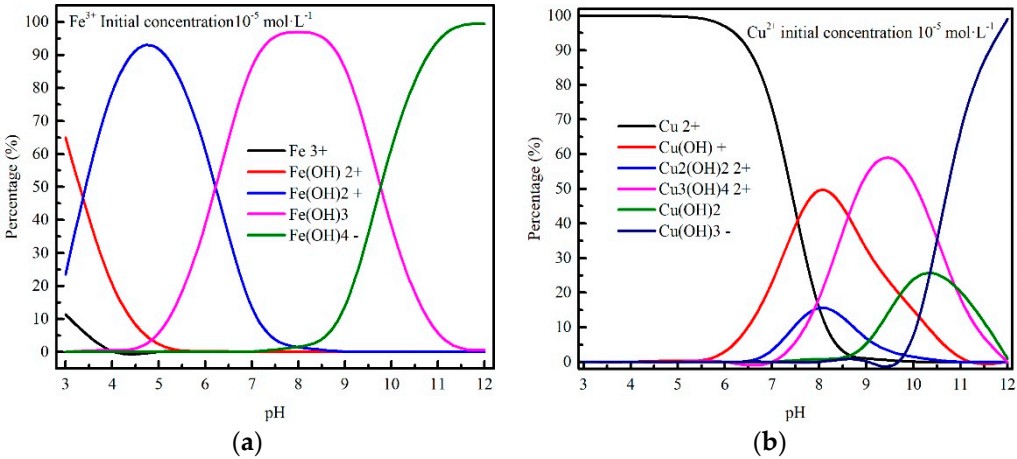

**Figure 5.** Species of (**a**) ferric ions in solution and (**b**) copper ions in solution.

### 3.6. Electrochemical Study

Many studies have strongly supported the flotation of sulfide minerals in electrochemical processing. Figure 6a shows the Tafel plots of the chalcopyrite electrode in 0 and 50 mg·L$^{-1}$ CMC solution. The inhibition efficiency (IE) was calculated from the corrosion current density as follows [38]:

$$IE\% = \frac{I^*_{corr} - I_{corr}}{I^*_{corr}} \times 100$$

where $I^*_{corr}$ and $I_{corr}$ are the corrosion current densities of chalcopyrite in 0 and 50 mg·L$^{-1}$ CMC, respectively. The corrosion current densities were $1.3 \times 10^{-5}$ and $9.2 \times 10^{-7}$ A·cm$^{-2}$ in CMC = 0 mg·L$^{-1}$ and CMC = 50 mg·L$^{-1}$, respectively. The IE% of CMC was 92.9%, which means that most of the reaction sites on chalcopyrite were inhibited by CMC.

Figure 6b shows the cyclic voltammetry results of the chalcopyrite electrode in 0 and 50 mg·L$^{-1}$ CMC solution. Two anodic peaks appeared—one at 0.3 V (peak A1) and the other at 0.6 V (peak A2). Cathodic peaks were observed at 0.08 V (peak C1) and −0.4 V (peak C2). In previous studies, the A1 peak was attributed to reactions (1) and (2) below [39,40]. Reaction (1) generates an intermediate product CuS* along with Fe$^{3+}$, which undergoes a precipitation reaction with CMC. The surface-inhibitor film on chalcopyrite reduced the current density of the oxidation peak A1. As the potential increased, the intermediate product CuS* and inner chalcopyrite were oxidized via reactions (3) and (4). The newly generated Fe$^{3+}$ and Cu$^{2+}$ ions further precipitated with CMC, causing a sharp decline in the current density from peak A2. The cathodic peaks C1 and C2 may be associated with reactions (5) and (6), respectively. Since the oxidation reactions inducing peaks A1 and A2 were inhibited by CMC, the current densities at peaks C1 and peak C2 were low. The CMC molecule is hydrophilic and the chalcopyrite surface is covered with CMC. The inhibition mechanism of CMC chemisorption on chalcopyrite flotation is shown in Figure 7a.

$$CuFeS_2 = CuS^* + Fe^{3+} + S + 3e^- \tag{1}$$

$$S + H_2O = SO_4{}^{2-} + e^- + H^+ \tag{2}$$

$$CuS^* = Cu^{2+} + S + 2e^- \tag{3}$$

$$CuFeS_2 = Cu^{2+} + Fe^{3+} + 2S + 5e^- \tag{4}$$

$$Fe^{3+} + e^- = Fe^{2+} \tag{5}$$

$$S + 2e^- = S^{2-} \tag{6}$$

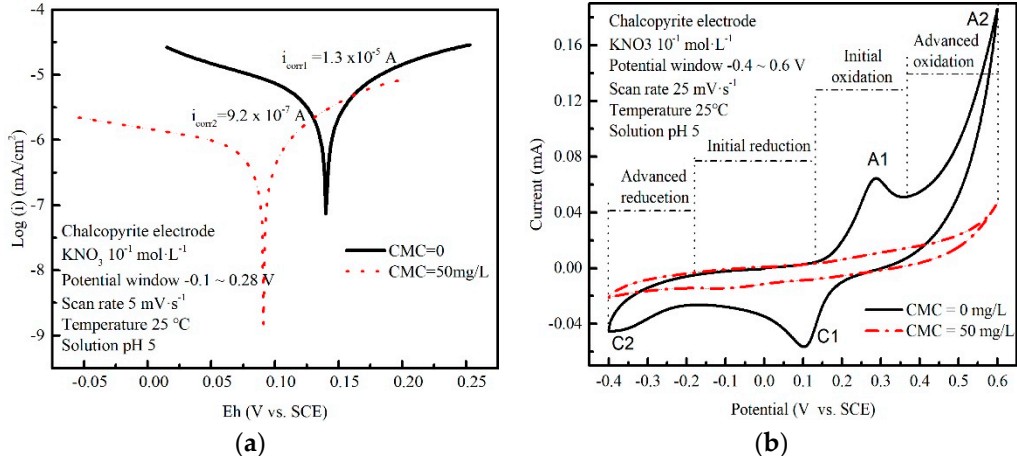

**Figure 6.** (**a**) Tafel plots and (**b**) cyclic voltammetry diagrams of chalcopyrite in the absence (black) and presence (red) of CMC.

### 3.7. FTIR Analysis

Infrared spectroscopy can reveal the interaction between chalcopyrite and CMC. The FTIR spectra of chalcopyrite in the presence and absence of CMC are shown in Figure 7b. The main CMC groups are –OH (3431, 1126, and 1047 $cm^{-1}$), –COOH (3431 and 1607 $cm^{-1}$), and –$CH_2$– (2916, 1331, and 1425 $cm^{-1}$) [22,39]. The broad band of chalcopyrite centered around 3430 $cm^{-1}$ is probably attributable to the –OH of Fe–OH. The peaks at 475 and 645 $cm^{-1}$ are the bending vibrations of the hydroxyl groups of lepidocrocite, reported as the main oxide of pyrite [40,41]. After the CMC treatment, the spectrum of chalcopyrite exhibited many new peaks. The stretching vibration peaks of –$CH_2$–OH (1047 $cm^{-1}$) and –C(HR)–OH (1126 $cm^{-1}$) were merged into a single narrow peak (1134 $cm^{-1}$). The absorption peak of the carboxyl groups shifted to the right (from 1607 to 1636 $cm^{-1}$). The in-plane bending vibrations of –C–H (at 1131 and 1425 $cm^{-1}$) were greatly weakened. Groups containing lone electron pairs (C–OH and COOH) participated in the chelation of $Cu^{2+}$ and $Fe^{3+}$.

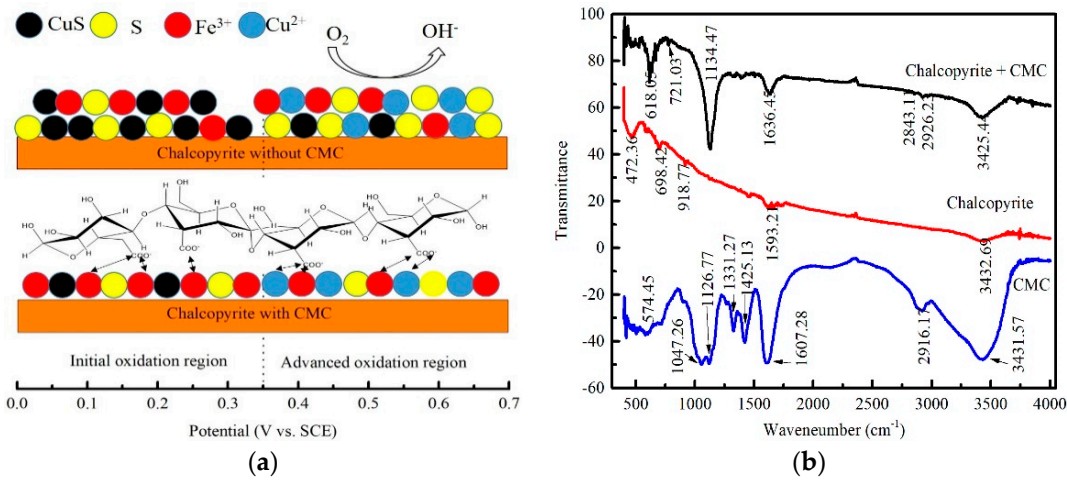

**Figure 7.** (**a**) Mechanism of the CMC-inhibited electrochemical reaction of chalcopyrite; (**b**) FTIR spectra of chalcopyrite in the absence (black) and presence (red) of CMC, the CMC spectrum is shown in blue.

### 3.8. XPS Measurements

Figure 8 illustrates the XPS spectra of Cu 2p, Fe 2p, S 2p, and C 1s in the absence and presence of CMC. In the XPS spectrum of Cu 2p (Figure 8a), the binding energies of Cu 2p and Cu $2p_2$ in the absence of CMC are centered at approximately 932.4 eV and 952.2 eV, respectively [42,43]. The intensities of

these peaks obviously declined after CMC addition because the chalcopyrite surface was covered with CMC molecules. The Fe on the chalcopyrite surface existed mainly in the forms of $CuFeS_2$ and FeO–OH, as shown in Figure 8b. The existence of $Fe^{3+}$ sites in chalcopyrite was confirmed around 719 eV [44,45]. After the CMC treatment, the peak at 719 eV disappeared, indicating that $Fe^{3+}$ had reacted with the CMC. $SO_4^{2-}$ was abundant on the surface and no $S^0$ was found (Figure 8c). This suggests that $S^0$ was unstable and quickly oxidized to $SO_4^{2-}$. A CuS signal was found at approximately 162 eV. Before the CMC treatment, some carbohydrate (C–O) was present on the chalcopyrite surface, and C=O was absent (Figure 8d). After treatment with CMC, large amounts of C–O and C=O were found on the surface. These results demonstrate the efficacy of carboxylic-containing carbohydrate as a chalcopyrite depressant.

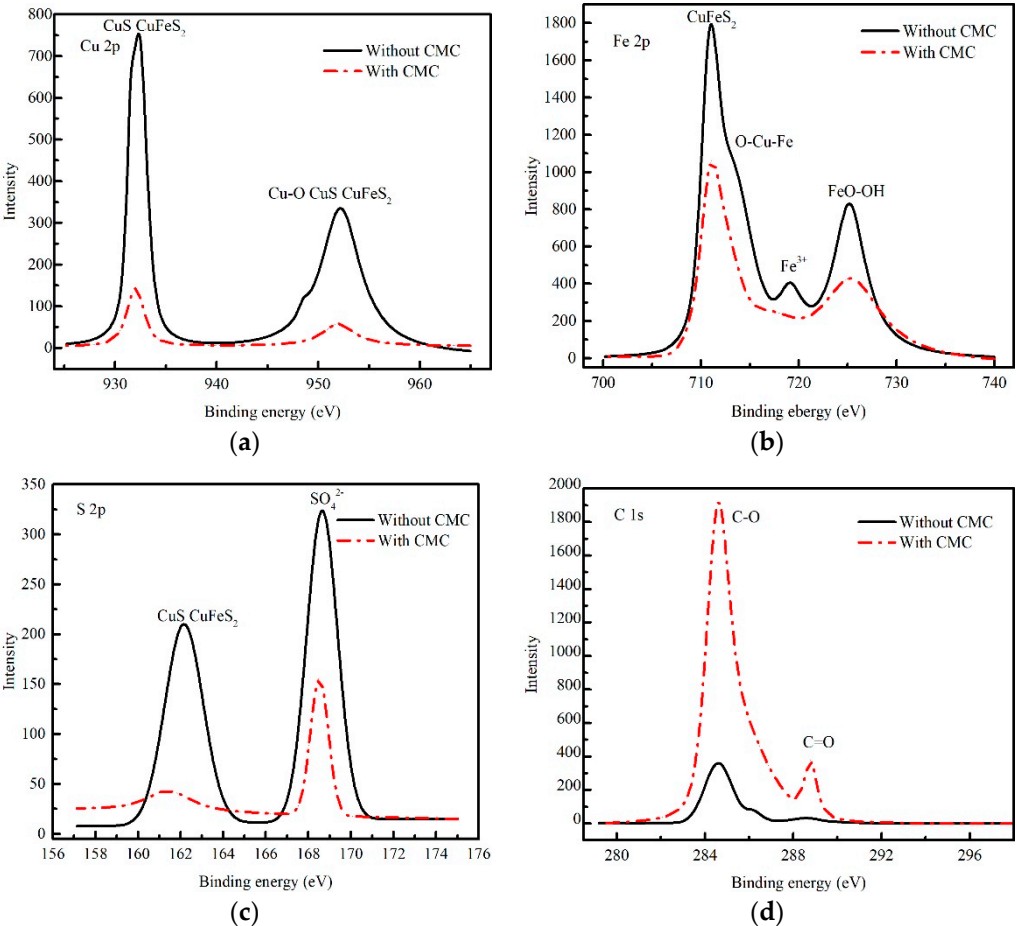

**Figure 8.** XPS analyses of chalcopyrite elements before (black) and after (red) treatment with CMC. (**a**) Cu 2p; (**b**) Fe 2p; (**c**) S 2p; (**d**) C 1s.

## 4. Conclusions

Chalcopyrite was entirely inhibited by 50 mg·$L^{-1}$ CMC in the 4–6 pH range, with minor effects on molybdenite flotation. The efficacy of CMC as a chalcopyrite depressant increased with increasing DS and $M_w$. Low DS and high $M_w$ were detrimental to molybdenite flotation. The separation flotation of artificial mixed minerals yielded Mo with a grade of 44.7%. CMC adsorption was favored by a positive zeta potential and hindered by carboxyl group protonation. CMC inhibited 92.9% of the electrochemical reaction sites on the chalcopyrite surface, and the production of hydrophobic substances was very low. The inhibition was mostly attributed to $Fe^{3+}$ on the chalcopyrite surface and to carboxyl groups in the CMC molecular structure.

**Author Contributions:** Conceptualization, H.Y. and G.C.; Methodology, X.Q. and G.C.; Software, X.Q.; Validation, X.Q., H.Y., G.C. and W.L.; Investigation, X.Q. and G.C.; Writing-Original Draft Preparation, X.Q., G.C. and H.Y.; Writing-Review & Editing, X.Q., H.Y. and G.C.

**Funding:** This research was funded by the National Natural Science Foundation of China, U1608254; State Key Laboratory of Comprehensive Utilization of Low-Grade Refractory Gold Ores, ZJKY2017(B)KFJJ01, ZJKY2017(B)KFJJ02; the Fundamental Research Funds for the Central Universities, N172504022; The National Key Research and Development Program of China (Project No. 2018YFC1902003).

**Acknowledgments:** This research was supported by Northeastern University Testing Center and the Zijin Mining Group. The authors would also like to acknowledge Jianing Xu, Desheng Zhang, Hefei Zhao, Haijun Li, Linlin Tong, and Zhenan Jin for technical and laboratory assistance.

**Conflicts of Interest:** The authors declare no conflict of interest.

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
