# Peer review of "An Alternative Depressant of Chalcopyrite in Cu–Mo Differential Flotation and Its Interaction Mechanism"

_minerals, doi:10.3390/min9010001_

Reviewer 1 Report

2.1 Materials In the XRD patterns (Fig 1a), please label the corresponding peaks. The total elemental contents in the XRF results do not equal to 100%. What are the other components? 2.2 Micro-flotation experiment Please illustrate when the suspension pH was recorded and what is the value reported in the manuscript. What is the particle size of the solids used for the flotation tests? 2.3 Adsorption tests What is the particle size of the solids used for the adsorption tests? How the authors guaranteed that 20 min was long enough for adsorption equilibrium to be achieved? 2.4 Zeta potential measurements What is the micro-ISA mill? Whether the authors can provide some information about the manufacture? Why the KCl concentration was selected at 0.1 M. It seems to be very high concentration. 3. Results and discussion In Figure 2(b), % need to be added into the axial titles. In Figure 4(b), the impacts of IEP on CMC adsorption needs to be reconsidered. As a chemisorption, the authors need to figure out why the physical characteristics (e.g., IEP) played that significant role in CMC adsorption. In Figure 5, instead of solely considering hydrolyzed species, please include the sulfide/sulfate species into calculation. Equation (2) showed that these species were formed after surface oxidization. Please make sure Figure 5 was mentioned in the text and in some places Figure 6 was incorrectly referred as Figure 5. Is there any supports for the expression “The broad band of chalcopyrite centered around 3430 cm−1 is probably attributable to hydrogen bonding with the hydroxyl groups of Fe–OH.” Whether the authors have any idea about what are the compounds of Fe3+ on the chalcopyrite surface? General Conclusion: The authors did a systematic study regarding the interaction between CMC and chalcopyrite, which explained the depression effects of CMC on chalcopyrite in the molybdenite/chalcopyrite flotation system. The study is meaningful and scientific. However, regarding the interaction mechanism, the author needs to clarify the following aspects: (1) If Fe are the active sites for the CMC adsorption, why the maximum depression occurred in pH 4-6; (2) In Figure 7(a), both Cu2+ and Fe3+ were marked as active sites for adsorption. However, the test results indicated that Fe3+ are the dominant active sites; (3) What is the function of the lateral interaction between CMC molecules in the adsorption.

Author Response

We tried our best to improve the manuscript and made some changes in the manuscript. These changes will not influence the content and framework of the paper.

We appreciate for your warm work earnestly, and hope that the correction will meet with approval.

Once again, thank you very much for your comments and suggestions.

Reviewer 2 Report

This is a well-organized flotation research paper indicating depression mechanisms of CMC for the chalcopyrite in Cu-Mo flotation system.

I think this study contributes to the literature and it can be accepted after minor revision

1.     There is slight typographical errors appear throughout the paper

2.     Abstract can be rewritten more fluently and precisely for the readers to understand the core units of the study.

3.     Introduction; are there any other potential selective depressants such as ATDT, quebracho and tannin for copper-molybdenum flotation separation?

4.     Figures legends were too confusing for the readers to follow the results

5.     P2 Section 2.1, Line 3 from the top; why the authors preferred to use pure minerals in the study since the real flotation system includes copper ores containing Mo? The readers can understand better if this is clear in the text.

6.     P3; What does inhibitory depressant mean?

7.     P3 Section 3.3  is there any reference for the CMC might flocculate the mineral particles?

8.     Does Figure 3(a) shows the 47% grade of the Mo?

9.     P4 Line 8 from top; is there any examples to many previous studies?

10.  P4, Section 3.5 Line 4 from top; from the figure 4 (a) pH increased from 3 to 11?

11.  Figures numbers are confusing, i.e., Figure 5(b) should be Figure 6(b) in P5 Line 5 from below?

12.  In Figure 6, there is some typographical errors such as “reduction”.

13.  P4, Section 3.5 Line 5 from below

14.  In Figure 6 readers can follow the peaks A1, A2, C1 and C2 mentioned in the text better if they are illustrated in the figure?

15.  What does Figure 7(a) indicate regarding ions of Fe3+ ?

Author Response

(The authors gave the same response as above.)

Reviewer 3 Report

I have reviewed the manuscript titled "An alternative depressant of chalcopyrite..." considered for publication in Minerals, and hereby express my opinion of the paper.

- The manuscript is exceptionally well written and easy to understand (good level of English language). The structure is logical, easy to follow, and I do not find any flaws in this regard. I also note that figures are clear, and when relevant error bars and relevant information on experimental conditions are included in figures. 

- The theory/introduction part could surely be expanded by additional references, as CMC is well established as depressant in various contexts. I also understand, however, the need to keep this part compact and surely 24 references should be sufficient to introduce the topic and current state-of-art.

 - A wide variety of experimental methods have been utilised, and as far as I am concerned they give very good and scientifically sound groundwork for the conclusions drawn as result of the study. I only have a few minor details to comment on regarding scientific method and how the work is presented, as listed below:

- I do not fully follow the figure references in 3.6 Electrochemical study, particularly the references to Figure 6 and 5. It is said that Figure 5 shows cyclic voltametry results, while the actual figure does not. I think this might be some simple figure referencing mistake, or that figures have been reordered or replaced in the writing process. Please check or clarify. Also there are references to peaks A1, A2, C1, C2 which are not depicted in any of the figures. As I am not familiar with the details of these particular methods, I apologise in advance if it is me not understanding.

- It might be good to express dosages of depressant and collector as g/ton of mineral or ore somewhere in the paper. Expressing as mg/L will not be clear to many readers. Calculating with the solids concentration used I come up to 500-600g/ton CMC which is a bit high but not unreasonable. SEX dosages expressed as mol/L is even more difficult to relate to.

- In the materials or micro-flotation section (2.1 or 2.2) you should present some specifics on the collector used and from where it was obtained. Now there is only a brief mention in some of the figures.  

The paper can more or less be published as is. With some minor clarifications made as recommended above, I do not hesitate to recommend the paper for publication.

Author Response

(The authors gave the same response as above.)
